# Cancer risk in individuals with intellectual disability in Sweden: A population-based cohort study

Qianwei Liu[1]*, Hans-Olov Adami[2,3,4], Abraham Reichenberg[5,6], Alexander Kolevzon[5,6,7], Fang Fang[1‡], Sven Sandin[2,5,6‡]

**1** Institute of Environmental Medicine, Karolinska Institutet, Stockholm, Sweden, **2** Department of Medical Epidemiology and Biostatistics, Karolinska Institutet, Stockholm, Sweden, **3** Institute of Health and Society, University of Oslo, Oslo, Norway, **4** Department of Epidemiology, Harvard T.H. Chan School of Public Health, Boston, Massachusetts, United States of America, **5** Department of Psychiatry, Ichan School of Medicine at Mount Sinai, New York, New York, United States of America, **6** Seaver Autism Center for Research and Treatment, Ichan School of Medicine at Mount Sinai, New York, New York, United States of America, **7** Department of Pediatrics, Icahn School of Medicine at Mount Sinai, New York, New York, United States of America

‡ These authors are joint senior authors on this work.
* qianwei.liu@ki.se

**Data Availability Statement:** Data cannot be shared publicly owing to restrictions by law. Data can be requested through the Statistics Sweden (information@scb.se) and the Swedish National

## Abstract

### Background

A knowledge gap exists about the risk of cancer in individuals with intellectual disability (ID). The primary aim of this study was to estimate the cancer risk among individuals with ID compared to individuals without ID.

### Methods and findings

We conducted a population-based cohort study of all children live-born in Sweden between 1974 and 2013 and whose mothers were born in a Nordic country. All individuals were followed from birth until cancer diagnosis, emigration, death, or 31 December 2016 (up to age 43 years), whichever came first. Incident cancers were identified from the Swedish Cancer Register. We fitted Cox regression models to calculate hazard ratios (HRs) and 95% confidence intervals (CIs) as measures of cancer risk in relation to ID after adjusting for several potential confounders. We analyzed ID by severity, as well as idiopathic ID and syndromic ID separately. We performed a sibling comparison to investigate familial confounding. The study cohort included a total of 3,531,305 individuals, including 27,956 (0.8%) individuals diagnosed with ID. Compared with the reference group (individuals without ID and without a full sibling with ID), individuals with ID were in general more likely to be male. The median follow-up time was 8.9 and 23.0 years for individuals with ID and individuals without ID, respectively. A total of 188 cancer cases were identified among individuals with ID (incidence rate [IR], 62 per 1,000 person-years), and 24,960 among individuals in the reference group (IR, 31 per 1,000 person-years). A statistically significantly increased risk was observed for any cancer (HR 1.57, 95% CI 1.35–1.82; P < 0.001), as well as for several

Board of Health and Welfare
(registerservice@socialstyrelsen.se) after approval
by the Ethics Committees.

**Funding:** The study was supported by grants from
the European Union (H2020-SC1: PM04-2016,
grant for SS). This study was also supported by the
Swedish Cancer Society (No. 20 0846 PjF, grant for
FF), the Swedish Research Council for Health,
Working Life and Welfare (No. 2017-00531, grant
for FF), the China Scholarship Council (No.
201700260291, grant for QL) and the Karolinska
Institute (Senior Researcher Award and Strategic
Research Area in Epidemiology, grant for FF). The
funding source had no role in study design, data
collection, data analysis, data interpretation, writing
of the scientific article, or the decision to submit
the paper for publication.

**Competing interests:** I have read the journal's
policy and the authors of this manuscript have the
following competing interests: AK reported
research support from AMO pharma and consult to
ovid, acadia, alkermes, jaguar, and ritrova. Other
authors declared no competing interests.

**Abbreviations:** ALL, acute lymphoid leukemia;
AML, acute myeloid leukemia; CI, confidence
interval; CNS, central nervous system; HR, hazard
ratio; ID, intellectual disability; IQ, intelligence
quotient; IR, incidence rate.

cancer types, including cancers of the esophagus (HR 28.4, 95% CI 6.2–130.6; $P < 0.001$), stomach (HR 6.1, 95% CI 1.5–24.9; $P = 0.013$), small intestine (HR 12.0, 95% CI 2.9–50.1; $P < 0.001$), colon (HR 2.0, 95% CI 1.0–4.1; $P = 0.045$), pancreas (HR 6.0, 95% CI 1.5–24.8; $P = 0.013$), uterus (HR 11.7, 95% CI 1.5–90.7; $P = 0.019$), kidney (HR 4.4, 95% CI 2.0–9.8; $P < 0.001$), central nervous system (HR 2.7, 95% CI 2.0–3.7; $P < 0.001$), and other or unspecified sites (HR 4.8, 95% CI 1.8–12.9; $P = 0.002$), as well as acute lymphoid leukemia (HR 2.4, 95% CI 1.3–4.4; $P = 0.003$) and acute myeloid leukemia (HR 3.0, 95% CI 1.4–6.4; $P = 0.004$). Cancer risk was not modified by ID severity or sex but was higher for syndromic ID. The sibling comparison showed little support for familial confounding. The main study limitations were the limited statistical power for the analyses of specific cancer types, and the potential for underestimation of the studied associations (e.g., due to potential underdetection or delayed diagnosis of cancer among individuals with ID).

## Conclusions

In this study, we found that individuals with ID showed an increased risk of any cancer, as well as of several specific cancer types. These findings suggest that extended surveillance and early intervention for cancer among individuals with ID are warranted.

---

## Author summary

### Why was this study done?

- Earlier clinical studies have suggested an increased risk of some specific types of cancer among individuals with genetic syndromes known to be associated with intellectual disability (ID), such as Down syndrome, fragile X syndrome, and tuberous sclerosis complex.

- Only a few population-based studies have examined the association between ID and cancer risk, and these studies are, in general, hampered by limited sample size and lack of adjustment for potential confounders, including familial confounding.

### What did the researchers do and find?

- We conducted a population-based cohort study including more than 3.5 million Swedish children born from 1974 to 2013, to examine the association between ID and cancer.

- We observed an increased risk of any cancer, as well as of several specific cancer types, among individuals with ID.

### What do these findings mean?

- Considering barriers to accessing healthcare and the vulnerability of individuals with ID, our findings could be important for surveillance and early intervention for cancer among individuals with ID.

- Future studies are needed to assess the risk of some specific types of cancer, especially cancers mostly diagnosed at later age.

## Introduction

Intellectual disability (ID) is a lifelong impairment of cognition and adaptive behavior that emerges in childhood [1], affects around 1% of the world population [2], and is associated with increased morbidity and mortality [3–5]. For instance, individuals with ID generally have an abnormal level of intelligence quotient (IQ) (under 70) and deficiency in at least 2 adaptive behaviors in environment and social milieu [6]. The underlying causes of ID are heterogenous, including chromosomal abnormality, gene mutation, environmental factors, and prenatal factors [7]. The exact cause, however, is not identifiable for most individuals with ID [8].

Cancer is one of the major causes of death before age 14 years and affects around 3.4% of males and 5.6% of females before the age of 49 years in the US [9]. Emerging evidence has suggested a plausible link between ID and cancer through several potential mechanisms. One possibility is that the chromosomal abnormalities or genetic mutations causing ID, especially syndromic ID, might also contribute to oncogenesis [10–12]. Another possibility is that individuals with ID may be more likely to be exposed to potential risk factors for cancer, such as unhealthy lifestyles including less optimal diets and lack of physical activity, which might also contribute to the initiation or development of some cancers [13,14].

However, population-based studies examining the association between ID and cancer are largely missing. Earlier clinical studies have suggested an increased risk of some specific types of cancer among individuals with genetic syndromes known to be associated with ID, such as Down syndrome, fragile X syndrome, and tuberous sclerosis complex [15–17]. There is, however, no consensus thus far about the risk of cancer in general among individuals with syndromic ID [3,18–22]. In addition, as previous studies have predominantly focused on syndromic ID, little is known about cancer risk among individuals with any ID, especially idiopathic ID.

Among the few studies of cancer risk in individuals with any ID, most used cancer death or specific cancer types as the outcome of interest [23–27]. Only a few population-based studies have assessed the risk of any cancer in relation to ID [28,29]. These studies, however, had some methodological limitations including small sample size and lack of adjustment for familial and birth characteristics, thereby limiting study validity and the possibility of studying rare cancers [28,29]. Close to 20% of all new cancers today are rare (<15 per 100,000 person-years) [30], and this proportion is even higher among children. To test the hypothesis that there is an association between ID and increased risk of cancer, population-based studies with large sample sizes and longitudinal follow-up are required.

To this end, we performed a large population-based cohort study to assess the association between ID and risk of cancer, using detailed information on more than 3.5 million individuals in Sweden. We examined the association by ID severity and type (i.e., idiopathic or syndromic), adjusted for several potential confounders including birth characteristics, and performed a sibling comparison to adjust for potential familial confounding.

## Methods

### Study population

We used nationwide data from Sweden made available via the European Union's Horizon 2020 research and innovation program RECAP preterm (Research on European Children and

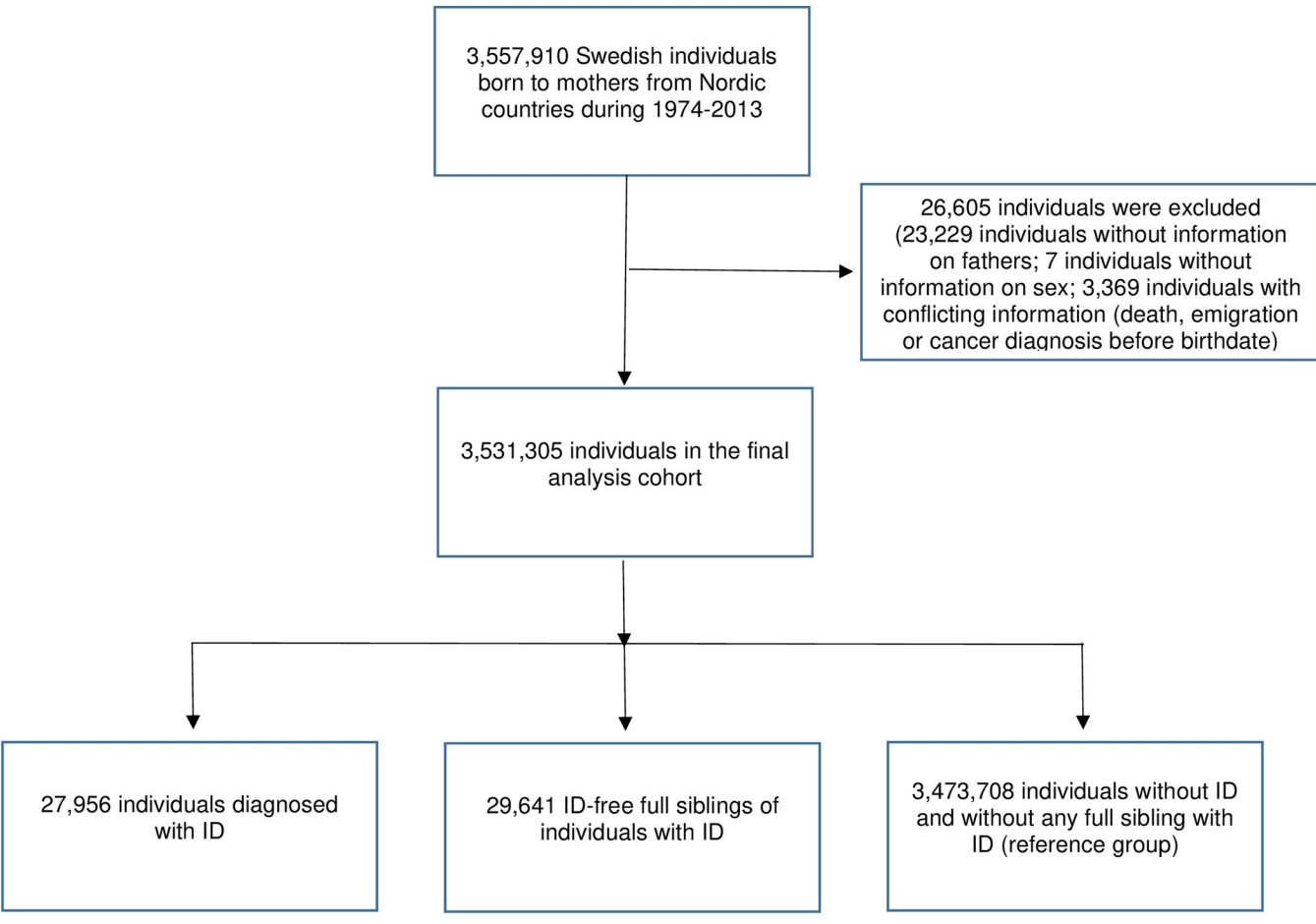

**Fig 1. Flow chart describing inclusion and exclusion of study participants.** ID, intellectual disability.

Adults Born Preterm; https://www.recap-preterm.eu). The study population consisted of all children live-born in Sweden between 1974 and 2013 and whose mothers were born in a Nordic country. The mothers of these individuals were identified through the Swedish Medical Birth Register, which covers 99% of all births in Sweden since 1973 [31]. Fathers were identified using the Swedish Multi-Generation Register [32].

We first identified a total of 3,557,910 individuals in the study cohort (Fig 1). Individuals with no information on father ($N = 23,229$), no information on sex ($N = 7$), or inconsistent information (emigration, death, or cancer diagnosis before birth, $N = 3,369$) were excluded from the analysis. The prespecified analysis plan is presented in S1 Text.

## Ascertainment of ID

ID cases were defined as individuals with a clinical diagnosis of ID during follow-up in the Swedish National Patient Register, which compiles hospital discharge records since 1964 and covers all inpatient discharges since 1987 and outpatient specialist care since 2001 [33]. In Sweden, primary healthcare services for children are provided by child welfare centers that support the health and development of children from infancy to 6 years of age, and nearly 100% of children participate in the follow-up visits [34]. As part of the Swedish child healthcare program, routine medical and developmental screening is regularly conducted for all neonates and

preschool children. A more detailed developmental assessment (motor, language, cognitive, and social development) is done at 2.5 and 4 years of age. Children with a suspected ID or other developmental abnormality are referred to a specialized team of pediatricians and child psychologists for further investigation.

We used the Swedish revisions of the International Classification of Diseases (ICD) codes in the National Patient Register to classify ID, ID severity (mild, moderate, severe, profound, other, or unspecified), and ID type (idiopathic or syndromic). In the study, syndromic ID was defined as individuals with ID and coexisting congenital malformations or chromosomal abnormalities, whereas idiopathic ID was defined as individuals with ID and without coexisting congenital malformations or chromosomal abnormalities. ICD codes for ID, ID severity, and ID type are listed in S1 Table.

## Ascertainment of cancer

Incident cancers were identified through linkage to the Swedish Cancer Register, using the Swedish 7th revision of the ICD codes. By law, all malignant tumors have been reported to the register since 1958, with a completeness close to 100% [35]. We first calculated the frequency of individual cancer types among individuals with and without ID (S2 Table). We then investigated "any cancer" as well as specific cancer types with at least 1 case among individuals with ID, including melanoma, non-melanoma skin cancer, Hodgkin lymphoma, non-Hodgkin lymphoma, acute myeloid leukemia (AML), acute lymphoid leukemia (ALL), and cancers of the salivary gland, esophagus, stomach, small intestine, colon, rectum, liver, pancreas, lung, breast, cervix, uterus, ovary, testis, kidney, eye, central nervous system (CNS), thyroid, other endocrine gland, bone, connective tissue, and other or unspecified sites (S3 Table). Individuals with more than 1 cancer type (1 individual with ID and 422 individuals without ID) were counted as a cancer case in the subgroup analysis of each applicable cancer type.

## Covariates

We collected information on variables from the Medical Birth Register, including sex (male/female), birth year, maternal age at delivery, maternal smoking during pregnancy (at first antenatal visit), multiple birth, gestational age at birth, birth weight, and Apgar score at 1 minute [31], to be studied as potential confounders or effect modifiers for the studied associations. Apgar score is a common measurement of the immediate health status of infants after delivery, based on skin complexion, heart rate, muscle tone, reflex irritability, and respiration effort. Apgar score has been shown to be associated with cognitive function as well as risk of childhood cancer [36,37]. We obtained information about paternal age at delivery through the Multi-Generation Register [32,38]. Maternal and paternal educational levels were collected from the Swedish Longitudinal Integration Database for Health Insurance and Labour Market Studies, which includes information on socioeconomic status [39]. Information on parental history of psychiatric disorders and cancer at delivery was collected at birth for the cohort participants from the National Patient Register and the Cancer Register, respectively (ICD codes listed in S1 and S3 Tables).

## Statistical analysis

The individuals were followed from birth until cancer diagnosis, emigration, death, or 31 December 2016 (up to age 43 years), whichever came first. We divided the individuals into 3 different groups. Individuals receiving a diagnosis of ID during follow-up were defined as the exposed group. As multiple factors clustering in families might be linked with both ID and cancer, we further defined another group consisting of ID-free full siblings of individuals with

ID. Finally, individuals without ID and without a full sibling with ID were defined as the reference group. If ID-free siblings or individuals without ID were later diagnosed with ID, they were censored from the sibling or reference group and moved to the exposed group on the date of diagnosis. Similarly, individuals without ID were censored from the reference group if a full sibling was later diagnosed with ID, and were moved to the group of ID-free full siblings of individuals with ID.

We calculated the unadjusted incidence rates (IRs) of cancer among individuals with ID and individuals in the reference group during follow-up, using the number of cases divided by accumulated person-years at risk. We used Cox regression models to estimate the relative risk of cancer among individuals with ID compared to the reference group by calculating hazard ratios (HRs) and associated 2-sided 95% Wald-type confidence intervals (CIs), corresponding to a statistical 2-sided test at the 5% level of significance. The HR is the instantaneous relative rate of cancer at any time during follow-up among individuals with ID, compared to individuals without ID [40]. Age at follow-up was used as the underlying time scale. For any cancer as well as for each cancer type, the analyses were performed in 2 models. In model 1, analyses were adjusted for sex (male/female) and birth year. To minimize the potential residual confounding due to a categorical representation of birth year, we fitted the models using natural cubic splines. In model 2, we additionally adjusted for maternal and paternal age at delivery as categorical covariates, and "inherited risk": presence of maternal and paternal history of psychiatric disorders (yes/no) or cancer (yes/no) at delivery. We repeated the analyses above in subgroups of sex and ID severity (mild, moderate, severe, profound, other, or unspecified), as well as for idiopathic ID and syndromic ID separately. To rule out familial confounding, i.e., confounding due to time-invariant factors common to full siblings, we performed a sibling comparison by fitting conditional Cox regression models to a dataset with differently exposed siblings with the family identifier (mother's and father's personal identification number) as strata. We did not adjust for multiplicity of statistical tests. Still, adopting a top-down approach, the main hypothesis of increased risk of any cancer among individuals with ID consisted of only 1 test.

## Supplementary and sensitivity analyses

We performed a sequence of supplementary and sensitivity analyses. First, as individuals with ID usually take an IQ test, it could be important to know whether IQ score impacts the association between ID and cancer risk. We derived IQ score from the ICD codes and examined the association between IQ and cancer risk (S1 Table). Second, to further explore the impact of birth characteristics, parental education, and maternal smoking during pregnancy as potential confounders of the association of ID with cancer, we, in separate models, adjusted for gestational age at birth (<37, 37–41, >41 weeks), birth weight (<2.5, 2.5–4, >4 kg), Apgar score at 1 minute, multiple birth (yes or no), maternal and paternal educational level (<9, 9–12, >12 years in school), and maternal smoking during pregnancy (yes or no). We also performed subgroup analyses by these variables to assess whether they could be effect modifiers of the studied association. Third, as the risk of both ID and cancer have been suggested to be increased in individuals born preterm [41,42], we repeated the main analysis among individuals born preterm. Fourth, as the association of ID with cancer might differ during childhood and early adulthood, we further restricted the main analysis to individuals aged ≤18 years. Fifth, to explore age-specific risks, we plotted survival curves of risk of cancer from Cox regression models adjusting for the covariates included in model 1. Sixth, as diagnostic criteria and screening strategies for ID and cancer might have changed during the study, we compared cancer risk among individuals born during 1974–1993 with those born during 1994–2013.

Seventh, to address influence of inherited cancer risk, we plotted cancer risk by the estimated heritability of cancer [43]. Eighth, analyzing only individual cancer types with at least 1 case among individuals with ID may introduce bias. In a sensitivity analysis, we studied cancer types by larger categories (i.e., by organ systems) regardless of the number of cases among individuals with ID. Ninth, to address the possibility of reverse causation (i.e., ID might be subsequent to cancer such as CNS tumors), we performed an additional analysis in which individuals with CNS cancer diagnosed within 5 years after the start of follow-up were excluded.

SAS software version 9.4 was used for data management and statistical analyses. The SAS codes for data management and the main analysis are presented in S2 Text. The study was approved by the Regional Ethical Review Board in Stockholm, Sweden (Dnr: 2017/1875-31/1). This study is reported according to the Strengthening the Reporting of Observational Studies in Epidemiology (STROBE) guideline (S1 STROBE Checklist).

## Results

The study cohort included a total of 3,531,305 individuals, including 27,956 (0.8%) individuals diagnosed with ID (15,334 with mild ID, 2,683 with moderate ID, 1,078 with severe ID, 450 with profound ID, and 8,411 with unspecified or other ID) and their 29,641 ID-free siblings (Tables 1 and S4). Among individuals with ID, 9,878 had syndromic ID (35.3%), whereas 18,078 had idiopathic ID (64.7%). Compared with the reference group, individuals with ID were in general more likely to be male, with lower parental educational level, lower birth weight, lower Apgar score at 1 minute, and a higher prevalence of multiple birth, preterm birth, parental psychiatric history, and maternal smoking during pregnancy (Tables 1 and S4). Characteristics of ID by severity are described in S5 Table.

The median follow-up time was 8.9 and 23.0 years for individuals with ID and individuals without ID, respectively. A total of 188 cancer cases were identified among individuals with ID (IR, 62 per 1,000 person-years) and 24,960 among individuals in the reference group (IR, 31 per 1,000 person-years). Individuals with ID, compared with the reference group, had a higher risk of subsequent cancer in both model 1 (HR 1.58, 95% CI 1.36–1.83; $P < 0.001$) and model 2 (HR 1.57, 95% CI 1.35–1.82; $P < 0.001$) (Table 2). All HRs presented below are from model 2.

Statistically significant associations were observed for several cancer types, including cancer of the esophagus (HR 28.4, 95% CI 6.2–130.6; $P < 0.001$), stomach (HR 6.1, 95% CI 1.5–24.9; $P = 0.013$), small intestine (HR 12.0, 95% CI 2.9–50.1; $P < 0.001$), colon (HR 2.0, 95% CI 1.0–4.1; $P = 0.045$), pancreas (HR 6.0, 95% CI 1.5–24.8; $P = 0.013$), uterus (HR 11.7, 95% CI 1.5–90.7; $P = 0.019$), kidney (HR 4.4, 95% CI 2.0–9.8; $P < 0.001$), CNS (HR 2.7, 95% CI 2.0–3.7; $P < 0.001$), and other or unspecified sites (HR 4.8, 95% CI 1.8–12.9; $P = 0.002$), as well as ALL (HR 2.4, 95% CI 1.3–4.4; $P = 0.003$) and AML (HR 3.0, 95% CI 1.4–6.4; $P = 0.004$) (Table 2).

The increased risk of any cancer did not vary between males and females or by ID severity (S1 and S2 Figs), but was higher among individuals with syndromic ID (Fig 2). For idiopathic ID, there was a statistically significantly increased risk of cancer of the esophagus, pancreas, and uterus and ALL, but not for any cancer. The sibling analysis yielded a similar point estimate of risk for any cancer in relation to ID as in the population analysis (Table 3).

### Supplementary and sensitivity analyses

First, we found no association between IQ level and cancer risk (S6 Table). Second, the association between ID and any cancer did not change by birth weight, Apgar score at 1 minute, gestational age at birth, multiple birth, parental education at delivery, or maternal smoking during pregnancy (S7 Table). Third, among individuals born preterm, we observed an

**Table 1. Characteristics of the cohort participants.**

| Variable | Number (%) of participants | | |
|---|---|---|---|
| | Individuals without ID (reference group) | Individuals with ID | ID-free full siblings of individuals with ID |
| **Number of individuals** | 3,473,708 | 27,956 | 29,641 |
| **Severity of ID** | | | |
| Mild ID | Not applicable | 15,334 | Not applicable |
| Moderate ID | Not applicable | 2,683 | Not applicable |
| Severe ID | Not applicable | 1,078 | Not applicable |
| Profound ID | Not applicable | 450 | Not applicable |
| Unspecified or other ID | Not applicable | 8,411 | Not applicable |
| **Sibship size (number of siblings)** | | | |
| 1 | 785,204 (22.6%) | 7,422 (26.5%) | 0 (0%) |
| 2 | 1,668,452 (48.0%) | 11,372 (40.7%) | 10,266 (34.6%) |
| ≥3 | 1,020,052 (29.4%) | 9,162 (32.8%) | 19,375 (65.4%) |
| **Male sex** | 1,783,489 (51.3%) | 16,221 (58.0%) | 15,091 (50.9%) |
| **Birth year** | | | |
| 1974–1983 | 895,736 (25.8%) | 5,311 (19.0%) | 5,732 (19.3%) |
| 1984–1993 | 977,135 (28.1%) | 10,326 (36.9%) | 11,430 (38.6%) |
| 1994–2003 | 771,253 (22.2%) | 8,857 (31.7%) | 8,452 (28.5%) |
| 2004–2013 | 829,584 (23.9%) | 3,462 (12.4%) | 4,027 (13.6%) |
| **Maternal age at delivery, years** | | | |
| <20 | 66,964 (1.9%) | 890 (3.2%) | 811 (2.7%) |
| 20–29 | 1,769,267 (50.9%) | 14,634 (52.3%) | 16,037 (54.1%) |
| 30–39 | 1,538,099 (44.3%) | 11,448 (41.0%) | 11,886 (40.1%) |
| ≥40 | 99,378 (2.9%) | 984 (3.5%) | 907 (3.1%) |
| **Paternal age at delivery, years** | | | |
| <20 | 15,143 (0.4%) | 202 (0.7%) | 135 (0.5%) |
| 20–29 | 1,261,215 (36.3%) | 10,534 (37.7%) | 11,181 (37.7%) |
| 30–39 | 1,855,087 (53.4%) | 13,622 (48.7%) | 14,714 (49.6%) |
| ≥40 | 342,263 (9.9%) | 3,598 (12.9%) | 3611 (12.2%) |
| **Maternal history of psychiatric disorder at delivery** | 205,069 (5.9%) | 3,094 (11.1%) | 2,514 (8.5%) |
| **Paternal history of psychiatric disorder at delivery** | 154,555 (4.5%) | 2,626 (9.4%) | 2,187 (7.4%) |
| **Maternal history of cancer at delivery** | 13,561 (0.4%) | 97 (0.3%) | 99 (0.3%) |
| **Paternal history of cancer at delivery** | 13,871 (0.4%) | 104 (0.4%) | 99 (0.3%) |

ID, intellectual disability.

increased risk of small intestine cancer (HR 89.7, 95% CI 5.6–1,439.8; $P = 0.002$) and ovarian cancer (HR 10.0, 95% CI 2.3–43.6; $P = 0.002$) among individuals with ID (S8 Table). Fourth, there was a higher risk of any childhood cancer among individuals with ID (HR 2.81, 95% CI 2.18–3.62; $P < 0.001$) (S9 Table). Fifth, by visual inspection (S3 Fig), the risk increases of cancer of the esophagus, stomach, small intestine, pancreas, and other or unspecific sites appear to be more pronounced among older than younger individuals. Sixth, we did not find a difference in risk of any cancer when separately analyzing individuals born 1974 to 1993 (HR 2.90, 95% CI 1.75–4.82; $P < 0.001$) and individuals born 1994 to 2013 (HR 2.74, 95% CI 1.92–3.91; $P < 0.001$) (S10 Table). Seventh, the association between ID and cancer was not related to cancer heritability (S4 Fig). Eighth, we observed statistically significant associations for cancers of the digestive system (HR 2.9, 95% CI 1.8–4.6; $P < 0.001$), urinary system (HR 2.8, 95% CI 1.3–6.3; $P = 0.01$), and CNS (HR 2.7, 95% CI 2.0–3.7; $P < 0.001$) as well as hematological malignancies

**Table 2. Incidence rates and hazard ratios of cancer among individuals with ID by cancer type, compared to the reference group.**

| Cancer type | Incidence rate of cancer (per 100,000 person-years) | | Hazard ratio (95% CI) | |
| --- | --- | --- | --- | --- |
| | Reference group | Individuals with ID | Model 1[a] | Model 2[b] |
| Any cancer | 31.42 | 62.49 | 1.58 (1.36–1.83) | 1.57 (1.35–1.82) |
| Salivary gland | 0.13 | 0.33 | 1.8 (0.2–12.8) | 1.8 (0.2–12.6) |
| Esophagus | 0.02 | 0.66 | 29.5 (6.5–133.4) | 28.4 (6.2–130.6) |
| Stomach | 0.08 | 0.66 | 6.0 (1.5–24.6) | 6.1 (1.5–24.9) |
| Small intestine | 0.04 | 0.66 | 11.6 (2.8–48.2) | 12.0 (2.9–50.1) |
| Colon | 0.91 | 3.64 | 2.0 (1.0–4.1) | 2.0 (1.0–4.1) |
| Rectum | 0.24 | 0.66 | 2.0 (0.5–8.0) | 2.0 (0.5–8.0) |
| Liver | 0.26 | 0.33 | 1.4 (0.2–9.6) | 1.4 (0.2–9.9) |
| Pancreas | 0.07 | 0.66 | 6.2 (1.5–25.4) | 6.0 (1.5–24.8) |
| Lung | 0.20 | 0.66 | 1.1 (0.2–8.0) | 1.1 (0.1–7.5) |
| Breast | 1.98 | 1.66 | 0.7 (0.3–1.6) | 0.7 (0.3–1.6) |
| Cervix | 1.54 | 1.66 | 0.8 (0.3–1.9) | 0.7 (0.3–1.8) |
| Uterus | 0.02 | 0.33 | 12.9 (1.7–98.4) | 11.7 (1.5–90.7) |
| Ovary | 0.48 | 1.32 | 2.2 (0.8–6.0) | 2.2 (0.8–5.9) |
| Testis | 2.86 | 6.29 | 1.3 (0.8–2.1) | 1.3 (0.8–2.1) |
| Kidney | 0.81 | 2.32 | 4.5 (2.0–10.2) | 4.4 (2.0–9.8) |
| Melanoma | 3.33 | 3.64 | 0.8 (0.4–1.4) | 0.8 (0.4–1.4) |
| Non-melanoma skin | 0.27 | 0.33 | 0.9 (0.1–6.6) | 0.9 (0.1–6.4) |
| Eye | 0.51 | 0.33 | 1.6 (0.2–11.2) | 1.5 (0.2–11.0) |
| CNS | 5.27 | 15.25 | 2.7 (2.0–3.7) | 2.7 (2.0–3.7) |
| Thyroid | 1.19 | 1.99 | 1.0 (0.4–2.4) | 1.0 (0.4–2.4) |
| Other endocrine gland | 1.87 | 3.97 | 1.5 (0.9–2.7) | 1.5 (0.9–2.7) |
| Bone | 0.69 | 0.66 | 0.8 (0.2–3.3) | 0.8 (0.2–3.3) |
| Connective tissue | 0.85 | 1.32 | 1.5 (0.6–4.0) | 1.5 (0.6–4.0) |
| Other or unspecified site | 0.21 | 1.32 | 5.1 (1.9–13.7) | 4.8 (1.8–12.9) |
| Hodgkin lymphoma | 1.61 | 1.99 | 0.9 (0.4–1.9) | 0.9 (0.4–1.9) |
| Non-Hodgkin lymphoma | 1.57 | 1.66 | 1.0 (0.4–2.4) | 1.0 (0.4–2.3) |
| ALL | 2.74 | 3.97 | 2.4 (1.3–4.4) | 2.4 (1.3–4.4) |
| AML | 0.79 | 2.32 | 3.0 (1.4–6.4) | 3.0 (1.4–6.4) |

[a]Analyses adjusted for birth year (as natural cubic splines) and sex.

[b]Analyses additionally adjusted for maternal and paternal age at delivery, maternal and paternal psychiatric disorder history at delivery, and maternal and paternal cancer history at delivery.

ALL, acute lymphoid leukemia; AML, acute myeloid leukemia; CI, confidence interval; CNS, to central nervous system; ID, intellectual disability.

(HR 1.5, 95% CI 1.1–2.2; $P = 0.02$) (S11 Table). Ninth, we observed a statistically significantly higher risk (HR 1.6, 95% CI 1.0–2.4; $P = 0.034$) of CNS cancer among individuals with ID after excluding CNS cancers diagnosed within 5 years after ID diagnosis.

## Discussion

In the largest population-based cohort study to date, to our knowledge, we observed a 1.6-fold increased risk of any cancer among individuals with ID, compared with individuals without ID, up to age 43 years. There was also an increased risk for several cancer types, including cancer of the esophagus, stomach, small intestine, colon, pancreas, uterus, kidney, CNS, and other or unspecified sites, as well as AML and ALL. The risk of any cancer was higher for syndromic ID and for childhood cancer, but did not vary by sex, ID severity, birth weight, Apgar score at

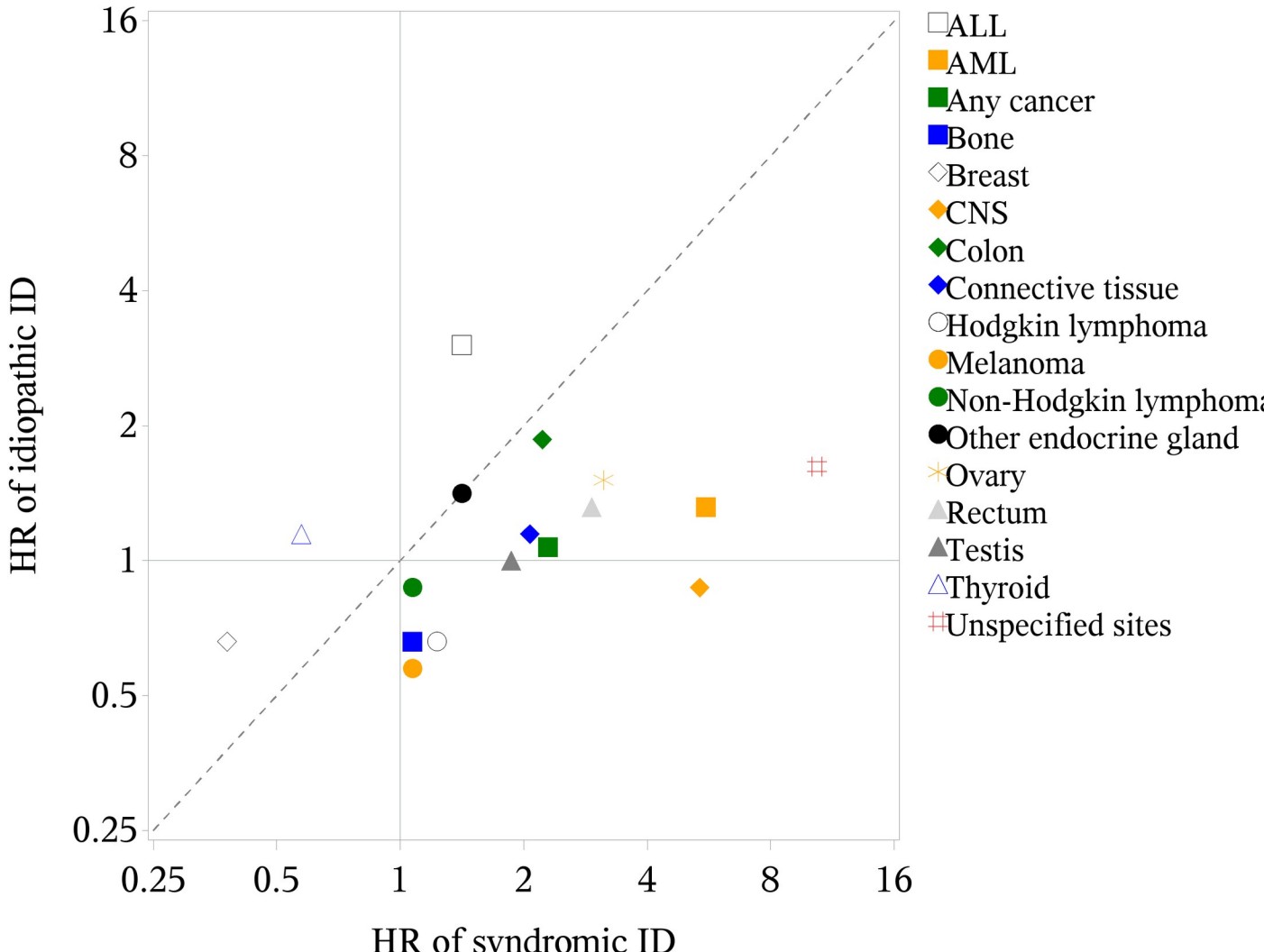

**Fig 2. HRs of cancer, by cancer type, among patients with idiopathic versus syndromic ID, compared to the reference group.** ALL, acute lymphoid leukemia; AML, acute myeloid leukemia; CI, confidence interval; CNS, central nervous system; HR, hazard ratio; ID, intellectual disability.

1 minute, gestational age at birth, parental education, maternal smoking during pregnancy, or calendar year of birth. The sibling comparison showed no support for familial confounding of the observed association.

### Strengths and weaknesses of this study

Strengths of our study include the population-based cohort design with large sample size, and the longitudinal and complete follow-up. Together with the independently collected information on ID and cancer, these strengths minimize the risk of selection and information biases. The adjustment for potential confounders and the sibling comparison could partly eliminate the concern that the observed associations were due to confounding (including familial confounding), which further strengthened the validity of our findings.

Our study also has limitations. Since individuals with ID usually have difficulty communicating signs and symptoms, they might experience delayed cancer diagnosis or underdiagnosis

**Table 3. Familial confounding: IRs (per 100,000 person-years) and HRs of cancer among individuals with ID, compared with their full siblings and the reference group—analyses restricted to individuals with at least 1 full sibling.**

| Cancer type | IR among individuals with ID and with at least 1 full sibling | Comparison with reference group | | Sibling comparison[a] | |
|---|---|---|---|---|---|
| | | IR among reference group with at least 1 full sibling | Model 2[b] HR (95% CI) | IR among ID-free full siblings of individuals with ID | Model 2[b] HR (95% CI) |
| Any cancer | 65.25 | 29.78 | 1.75 (1.47–2.08) | 42.53 | 1.59 (1.13–2.23) |
| Salivary gland | 0.46 | 0.13 | 2.6 (0.4–18.8) | 0.92 | — |
| Esophagus | 0.46 | 0.01 | 30.8 (3.7–253.8) | No case | — |
| Stomach | 0.46 | 0.07 | 4.1 (0.6–29.7) | No case | — |
| Small intestine | 0.46 | 0.03 | 11.4 (1.5–85.5) | No case | — |
| Colon | 4.15 | 0.86 | 3.0 (1.5–6.1) | 1.22 | — |
| Rectum | 0.92 | 0.20 | 3.3 (0.8–13.2) | No case | — |
| Liver | 0.46 | 0.25 | 2.0 (0.3–14.5) | 0.31 | — |
| Pancreas | 0.46 | 0.07 | 3.8 (0.5–27.9) | 0.31 | — |
| Lung | 0.46 | 0.20 | — | 0.31 | — |
| Breast | 1.38 | 1.57 | 0.7 (0.2–2.1) | 3.36 | — |
| Cervix | 0.46 | 1.31 | 0.2 (0.0–1.7) | 1.83 | — |
| Uterus | 0.46 | 0.01 | 24.0 (3.0–191.1) | No case | — |
| Ovary | 1.84 | 0.44 | 3.3 (1.2–9.0) | 0.61 | — |
| Testis | 6.46 | 2.64 | 1.4 (0.8–2.4) | 4.89 | 2.2 (0.2–25.1) |
| Kidney | 2.31 | 0.78 | 4.6 (1.7–12.4) | 0.61 | — |
| Melanoma | 2.31 | 3.04 | 0.5 (0.2–1.2) | 5.80 | 0.6 (0.1–2.4) |
| Non-melanoma skin | 0.46 | 0.23 | 1.4 (0.2–10.3) | 0.31 | — |
| Eye | No case | 0.49 | — | 0.31 | — |
| CNS | 18.47 | 5.26 | 3.4 (2.5–4.8) | 4.28 | 7.4 (2.0–27.5) |
| Thyroid | 2.31 | 1.10 | 1.1 (0.4–3.0) | 3.66 | 0.8 (0.1–6.4) |
| Other endocrine gland | 4.15 | 1.77 | 1.7 (0.9–3.3) | 3.97 | 0.9 (0.2–3.4) |
| Bone | 0.92 | 0.69 | 1.1 (0.3–4.6) | 0.61 | — |
| Connective tissue | 1.84 | 0.87 | 2.0 (0.7–5.4) | 0.92 | — |
| Other or unspecified site | 1.38 | 0.19 | 5.6 (1.8–17.7) | 0.31 | — |
| Hodgkin lymphoma | 2.31 | 1.58 | 1.0 (0.4–2.4) | 2.75 | — |
| Non-Hodgkin lymphoma | 1.84 | 1.52 | 1.1 (0.4–3.0) | 2.44 | 0.9 (0.1–14.0) |
| ALL | 4.15 | 2.81 | 2.8 (1.4–5.4) | 1.22 | — |
| AML | 2.77 | 0.81 | 3.7 (1.6–8.2) | 0.61 | — |

A dash indicates too few cancer cases to allow estimation.

[a]Analyses were stratified by family identifier using conditional Cox regression model.

[b]Analyses adjusted for birth year (as natural cubic splines), sex, maternal and paternal age at delivery, maternal and paternal psychiatric disorder history at delivery, and maternal and paternal cancer history at delivery.

ALL, acute lymphoid leukemia; AML, acute myeloid leukemia; CI, confidence interval; CNS, central nervous system; HR, hazard ratio; ID, intellectual disability; IR, incidence rate.

of cancer [44,45]. Although we had no data on cancer stage in the present study, we did notice a higher incidence of cancer of other or unspecified sites (i.e., cancer of unknown origin or diagnosed at metastatic stage) among individuals with ID, compared with others. Diagnostic

delay and underdiagnosis of cancer among individuals with ID could have led to underestimation of the studied associations. No access to ID diagnoses from primary care, and inclusion of outpatient diagnoses in the National Patient Register only from 2001 onwards, might potentially have led to underdetection of ID cases, especially mild ID. Because we focused on individuals with a clinical diagnosis of ID through the National Patient Register, ID cases generally not only had low IQ score, but also showed relatively severe symptoms, required extended care, or could not address daily tasks; therefore, the generalizability of our findings to individuals with mild cases needs further assessment. ID might be consequent to cancer such as CNS cancer. The increased risk of CNS cancer among individuals with ID even after excluding CNS cancers diagnosed within 5 years after ID diagnosis alleviated concern about reverse causation to a large extent. Restricting analysis to individual cancer types with at least 1 case among individuals with ID might have introduced bias to some extent. Interpretation of the cancer-type-specific results should therefore be cautious. Despite the large sample size, our study could only address cancer cases diagnosed up to 43 years of age and had limited power to assess the risk of cancers mostly diagnosed at later age, e.g., prostate cancer and lung cancer. Finally, we had no information on behavioral factors, such as diet and physical activity, which might confound or mediate the association of interest. However, as these factors are prone to cluster within families, the similar results of the sibling comparison should have partly alleviated such concern.

## Comparison with other studies

In line with previous research [46], we observed a higher prevalence of ID among males than females. One potential reason is that males have higher vulnerability to syndromes linked to the X chromosome, such as fragile X syndrome, which is related to higher risk of ID [47]. Another reason might be differential brain development due to different androgen exposure between males and females [48].

We observed an increased risk of cancer among individuals with ID, in contrast to previous studies that reported similar risk of cancer between ID patients and the general population [28,29]. The different results might arise because previous studies involved a wider age range, including older individuals, whereas we focused on childhood and early adulthood. Another potential reason for the different results might be the small sample size and the potential for random error in previous studies [28,29]. Patja et al. [28] studied 2,173 individuals with ID in a 30-year follow-up study of a Finnish population, whereas Sullivan et al. [29] studied 9,409 individuals with ID in a 19-year follow-up study of a population in Western Australia. With a total number of 27,956 individuals with ID and a follow-up of 43 years, the present study represents therefore, to our knowledge, the greatest effort of its kind to date. In addition, our study is the first to our knowledge to report the cancer risk of individuals with idiopathic ID. We observed no excess risk of any cancer, but increased risk of cancer of the esophagus, pancreas, and uterus, as well as ALL, among individuals with idiopathic ID, thereby expanding the previous knowledge base further.

Our study showed increased risks for colon and uterus cancers in females and CNS cancer in males among individuals with ID, which is in line with 1 previous study [29]. However, we did not observe statistically significantly increased risk of stomach cancer or decreased risk of prostate cancer among males [29], nor increased risk of gallbladder and thyroid cancers in general [28], among individuals with ID. The contrasting results may be partly due to the different age ranges of the studied participants in the different studies.

## Potential mechanisms

One potential mechanism for the observed association between ID and any cancer is that ID in some cases might be caused by multiple-system congenital anomalies affecting more than 1

organ [49]. For instance, Down syndrome, one of the leading causes of ID, is related to the presence of somatic mutations in *GATA1* and *JAK2*, which are associated with acute megakaryoblastic leukemia and ALL [10,11,50,51]. Tuberous sclerosis, another common cause of syndromic ID, is associated with increased risk of brain tumor due to mutations of *TSC1* and *TSC2* [52,53]. Lifestyle factors that differ between individuals with and without ID might also play a role. We observed an increased risk of cancer of the digestive system among individuals with ID. This could be associated with unbalanced diet as well as poor oral hygiene among individuals with ID [13,14,54,55]. To be noted, we found a markedly increased risk of esophagus cancer among individuals with ID, in line with previous findings [27], which might be related to a higher prevalence of gastroesophageal reflux disease, an established cause of esophagus cancer [56], in individuals with ID [57]. Moreover, overweight and physical inactivity among individuals with ID might also play a role in the association between ID and increased risk of several cancer types, including uterus and kidney cancer [58–60]. Our results suggest that individuals with ID who were born preterm had a markedly increased risk of ovarian cancer compared with the general population. Such risk elevation might be related to the higher prevalence of abnormal ovarian function and smaller ovary size among individuals with preterm birth [61], which might contribute to the effect of ID on ovarian cancer. As ID and cancer are both heterogeneous diseases with various etiologies, further studies are warranted to better understand the underlying mechanisms.

## Conclusion

There is an increased risk of any cancer, as well as of several specific cancer types, among individuals with ID. The associations could not be explained by shared genetics or other familial confounders of ID and cancer.

## Supporting information

**S1 Fig. Hazard ratios (HRs) of cancer among individuals with intellectual disability (ID) by sex, compared to the reference group.**
(PDF)

**S2 Fig. Hazard ratios (HRs) of cancer among individuals with ID by intellectual disability (ID) severity and by cancer type, compared to the reference group.**
(PDF)

**S3 Fig. Cancer risk overall (any cancer) and by cancer type among individuals with ID and in the reference group.**
(PDF)

**S4 Fig. Hazard ratios (HRs) of cancer among individuals with intellectual disability (ID) by heritability of cancer, compared to the reference group.**
(PDF)

**S1 STROBE Checklist. STROBE Statement—checklist of items that should be included in reports of observational studies.**
(PDF)

**S1 Table. ICD codes for intellectual disability (ID), severity of ID, ID type, intelligence quotient (IQ), psychiatric disorders, and congenital malformations and chromosomal abnormalities.**
(PDF)

**S2 Table. Numbers of cancer cases among individuals with ID and individuals without ID.**
(PDF)

**S3 Table. ICD codes for studied cancer types.**
(PDF)

**S4 Table. Birth characteristics, parental education, and maternal smoking during pregnancy of the cohort participants.**
(PDF)

**S5 Table. Characteristics of individuals with intellectual disability (ID) by severity of ID.**
(PDF)

**S6 Table. Association between IQ score and risk of cancer among individuals with intellectual disability, compared to the reference group, by cancer type.**
(PDF)

**S7 Table. Hazard ratios (HRs) with 95% confidence intervals (CIs) of any cancer among individuals with intellectual disability (ID), compared to the reference group, further separately adjusted for or stratified by maternal smoking during pregnancy, parental education at delivery, multiple birth, gestational age at birth, birth weight, and Apgar score at 1 minute.**
(PDF)

**S8 Table. Subgroup analyses of preterm birth: Incidence rates (IRs, per 100,000 person-years) and hazard ratios (HRs) with 95% confidence intervals (CIs) of cancer among individuals with intellectual disability (ID), compared to the reference group, restricting analyses to individuals who were born preterm.**
(PDF)

**S9 Table. Incidence rates (IRs, per 100,000 person-years) and hazard ratios (HRs) with 95% confidence intervals (CIs) of childhood cancer (age ≤ 18 years) among individuals with intellectual disability (ID) by cancer type, compared to the reference group.**
(PDF)

**S10 Table. Comparison between different calendar years: Incidence rates (IRs, per 100,000 person-years) and hazard ratios (HRs) with 95% confidence intervals (CIs) of cancer among individuals with intellectual disability (ID), compared to the reference group, for individuals born during 1994–2013 and those born during 1974–1993.**
(PDF)

**S11 Table. Incidence rates (IRs, per 100,000 person-years) and hazard ratios (HRs) with 95% confidence intervals (CIs) of cancer among individuals with intellectual disability (ID) by organ system, compared to the reference group.**
(PDF)

**S1 Text Analysis plan.**
(PDF)

**S2 Text SAS codes.**
(PDF)

## Author Contributions

**Conceptualization:** Qianwei Liu, Hans-Olov Adami, Fang Fang, Sven Sandin.

Data curation: Sven Sandin.

Formal analysis: Qianwei Liu, Sven Sandin.

Funding acquisition: Qianwei Liu, Fang Fang, Sven Sandin.

Investigation: Qianwei Liu, Hans-Olov Adami, Fang Fang, Sven Sandin.

Methodology: Qianwei Liu, Fang Fang, Sven Sandin.

Project administration: Fang Fang, Sven Sandin.

Resources: Fang Fang, Sven Sandin.

Supervision: Fang Fang, Sven Sandin.

Writing – original draft: Qianwei Liu.

Writing – review & editing: Hans-Olov Adami, Abraham Reichenberg, Alexander Kolevzon, Fang Fang, Sven Sandin.

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
