## [Editor Report · Decision Letter 0]

27 Apr 2021

Dear Dr Liu, 

Thank you for submitting your manuscript entitled "Cancer risk in individuals with intellectual disability" for consideration by PLOS Medicine.

Your manuscript has now been evaluated by the PLOS Medicine editorial staff and I am writing to let you know that we would like to send your submission out for external assessment.

However, before we can send your manuscript for assessment, we need you to complete your submission by providing the metadata that is required for full assessment. To this end, please login to Editorial Manager where you will find the paper in the 'Submissions Needing Revisions' folder on your homepage. Please click 'Revise Submission' from the Action Links and complete all additional questions in the submission questionnaire.

Please re-submit your manuscript within two working days, i.e. by Apr 29 2021 11:59PM.

Once your full submission is complete, your paper will undergo a series of checks in preparation for external assessment. 

Kind regards,

Richard Turner, PhD

rturner@plos.org

---

## [Decision Letter · Decision Letter 1]

11 Aug 2021

Dear Dr. Liu,

Thank you very much for submitting your manuscript "Cancer risk in individuals with intellectual disability" (PMEDICINE-D-21-01891R1) for consideration at PLOS Medicine. We apologize for the delay in sending you a response. 

Your paper was discussed with an academic editor with relevant expertise and sent to independent reviewers, including a statistical reviewer. The reviews are appended at the bottom of this email and any accompanying reviewer attachments can be seen via the link below:

[LINK]

In light of these reviews, we will not be able to accept the manuscript for publication in the journal in its current form, but we would like to invite you to submit a revised version that addresses the reviewers' and editors' comments fully. You will appreciate that we cannot make a decision about publication until we have seen the revised manuscript and your response, and we expect to seek re-review by one or more of the reviewers. 

We hope to receive your revised manuscript by Sep 01 2021 11:59PM. Please email us (plosmedicine@plos.org) if you have any questions or concerns.

Please let me know if you have any questions, and we look forward to receiving your revised manuscript. 

Sincerely,

Richard Turner, PhD

rturner@plos.org

Please adapt the title so that it contains a study descriptor after a colon, e.g., "Cancer risk in individuals with intellectual disability: A population-based cohort study".

Please adapt the abstract to a three-part structure. 

Rather than "Nordic mothers", please write "... of mothers born in ..." or similar, in the abstract and any other instances in the paper. 

In the abstract and throughout the paper, please quote p values alongside 95% CI, where available.

Please add a final sentence to the new "Methods and findings" subsection of your abstract, which should begin "Study limitations include ..." or similar and should quote 2-3 of the study's main limitations.

After the abstract, we will need to ask you to add a new and accessible "Author summary" section in non-identical prose. You may find it helpful to consult one or two recent research papers in PLOS Medicine to get a sense of the preferred style. 

Early in the Methods section, please state whether or not the study had a protocol or prespecified analysis plan, and if so attach the relevant document as a supplementary file, referred to in the text. 

Please highlight analyses that were not presecified. 

Please remove the information on funding and competing interests from the Acknowledgements section at the end of the main text. In the event of publication, this information will be included in the article metadata, via entries in the submission form. 

Throughout the text, please adapt reference call-outs to the following style: "... study rare cancers [11,12]." (noting the absence of spaces within the square brackets). 

Please abbreviate journal names consistently in your reference list.

Where appropriate, 6 author names should be listed rather than 3, followed by "et al.".

Please add a completed checklist for the most appropriate reporting guideline, e.g., STROBE, as a supplementary file, labelled "S1_STROBE_Checklist" or similar and referred to as such in the Methods section. 

In the checklist, please refer to individual items by section (e.g., "Methods") and paragraph number, not by line or page numbers as these generally change in the event of publication. 

Comments from the reviewers:

*** Reviewer #1: 

The study titled "Cancer risk in individuals with intellectual disability" is one of its kind in both the terms of topic selection as well as the large sample size that is evaluated in the study. It strives to create a association between intellectual disability and occurrence of any form of cancer. The study would have an impact in the guidelines of management of all forms of intellectual disability in the form of adding need of cancer screening in virtually all causes of intellectual disability. It might even cause a debate or ethical dilemma regarding the management and prognosis of intellectual disability. 

The strength of the study lies in the fact that:

1. The study is prospective in nature

2. The sample size is large

3. No population under study is left out: The authors claim that the children are screened and diagnosed early as well as the cancer registry is nearly encompasses all of the malignancies that might occur

4. Statistical analysis done robustly to remove certain confounding factors 

However certain facts cannot be overlooked while going through the study.

1. The "Introduction" section does not justify the caliber of the study or the journal. The section glances over the statistics regarding intellectual disability and malignancies. However, the causes of known and unknown causes of intellectual malignancies seem to be missing. The literature review seems to miss multiple similar papers. The scientifically proven causation of different forms of intellectual disabilities along with mutations attributed to malignancy development are not mentioned.

2. The "Methodology" section establishes the fact that cases under study are unlikely to be missed. However, not all forms of intellectual disabilities manifest at a young age. The authors mention about a mandatory wellness child examination at the age of four. They fail to mention where the examination occurs. For instance if it occurs in school, would the children at home, which is likely in intellectually disabled child due to logistic or other reasons, be missed? 

3. In the "Statistical analysis" section, does the comparison of IQ score with risk of cancer have any clinical significance?

4. The "Discussion" portion of the study does not compare multiple similar articles.

Ways to improve the article:

1. The authors need to perform literature review and update the "Introduction" and "Discussion" section.

2. If they have sufficient data from the registry, could they demonstrate the occurrences of proven malignancies to proven clinical diagnoses. e.g. prevalence of certain malignancies in Down syndrome. This would further validate the data.

3. Was the family history of malignancy collected? The manuscript does not explicitly mention this. If so, was the "end point" of malignancy detection in the intellectually disabled individuals the same as for family members.

4. The article needs overall editing of the language for it to be more comprehensible. Maybe technical terms could be better phrased so that it would find a wider audience.

5. Could the authors mention also about the diagnosed causes of intellectual disability according to prevalence?

6. The "Methodology" section could be overhauled to make it easier for the study to be reproduced.

7. Compare the malignancy trend with the trend in the country. Does it reflect the prevalence/incidence of malignancies as evidenced by other studies or cancer registry?

8. Present the data in a more palatable graphical representations instead of only tables. 

*** Reviewer #2: 

Overall the manuscript is written clearly and concisely, the topic is relevant, and the research extends existing knowledge. I would recommend this manuscript be published in PLOS medicine with some revisions. 

One of the most important revisions I suggest is to explain further what the hazard ratio statistic means, as this could easily be misunderstood. For example, when you say 'a 1.6-fold increased risk of any cancer' on page 12, tell the reader how they should interpret that. 

Spruance SL, Reid JE, Grace M, Samore M. Hazard ratio in clinical trials. Antimicrobial agents and chemotherapy. 2004 Aug;48(8):2787-92.

Sutradhar R, Austin PC. Relative rates not relative risks: addressing a widespread misinterpretation of hazard ratios. Annals of epidemiology. 2018 Jan 1;28(1):54-7.

More Minor recommendations:

 * You could change your title to include the fact that you did a population-cohort study in Sweden, this makes the life of future researchers easier when they come across your study. 

 * P. 5 - The first paragraph with the definition of ID is very clear. However, you could also give examples of the most common forms of ID to give the reader some context. 

 * P. 6 - Does 'Nordic mothers in Sweden' mean that everyone is of the same ethnicity or with is it simply related to citizenship? Was this something recorded in the data you had? This does not need to be something you add as a covariate into analyses necessarily, but it is interesting to know the diversity of the population you looked at. 

 * P. 7 - You could define what the Apgar score is and how it is relevant, as not everyone will know this. 

 * P. 7 - You said you 'obtained information about paternal age at delivery through Multi-Generation Register' - But, I wondered if you were successful at tracing all fathers? If not, did that impact some of your analysed sample sizes?

 * P. 9 - I have a few comments around the terms used for sex.

 * You include sex as a covariate in some analyses, but you do not state whether the levels for this are 2 (Female, Male) or 3 (Female, Male, Intersex). Please define this as you do for covariate such as severity of ID. 

 * Also, the terms sex and gender, though commonly used interchangeably in research, are seen as two distinct aspects of an individual's lived experience. 

 * Best current practice is to refer to sex at birth (Female, Male, Intersex/Undetermined) and gender identity (woman, man, non-binary, indigenous or other cultural gender minority [e.g., two-spirit], etc.,).

 * Slade T, Gross DP, Niwa L, McKillop AB, Guptill C. Sex and gender demographic questions: improving methodological quality, inclusivity, and ethical administration. International Journal of Social Research Methodology. 2020 Sep 12:1-2.

 * Most relevant to your article is sex at birth of course, but on pages 12 and 18 you refer to 'men and women'. It would be more appropriate for you to refer to male and female if you are interested in sex at birth rather than gender identity. 

 * P. 9 - Do you have a citation for 'the risk of both ID and cancer is increased in preterm born individuals,' ?

 * P. 10 - Do you have any code that you could share for your statistical analysis? 

 * P. 11 - The result you give for point (1) in your supplementary and sensitivity analyses section is very short and could be a bit clearer. In particular, what do you mean by 'similar'?

 * P. 12 - Be careful with using the word 'greatly', perhaps remove and just say that it 'did not vary by sex, ID severity' etc., as that is what you found. 

 * P. 12 - Similarly, you say 'not predominantly due to factor shared by siblings', but did you show that it was related to this at all? This relates back to your explanation of point 1 in sup/sensitivity analysis in results.

 * P. 12 - You mention that there is delay and under-diagnosis of cancer in people with IDs. This is an extremely impactful point that would be worth explaining a little further to the reader. 

 * Firstly, give a citation for your statement (e.g., Hogg and Tuffrey-Wijne, 2008, and say why is this could be the case (i.e. communication difficulties leading to lack of pain reporting - Hogg and Tuffrey-Wijne, 2008; Millard and de Knegt, 2019). 

 * Hogg J, Tuffrey‐Wijne I. Cancer and intellectual disability: a review of some key contextual issues. Journal of Applied Research in Intellectual Disabilities. 2008 Nov;21(6):509-18.

 * Millard SK, de Knegt NC. Cancer pain in people with intellectual disabilities: systematic review and survey of health care professionals. Journal of pain and symptom management. 2019 Dec 1;58(6):1081-99.

 * Related to this, do you know anything about the stage of cancer at diagnosis in your data? You may have been able to show that there is delayed diagnosis of cancer in people with IDs. 

 * P. 13 - You say that previous studies had smaller numbers and shorter durations, it may be useful to mention how large and long these studies where, and also in what populations. This could be done either in your introduction or here in the discussion. Having this clearly means that the reader understands the full scope of the study you've done, and if your study is so much bigger, you should display that. 

 * On a related point, how long is 'enough follow-up'?

 * Additionally, did you use any reporting guidelines? 

*** Reviewer #3: 

Please note that this is a statistical review - I cannot comment on the importance of findings, only the validity.

--

This paper assesses whether individuals with intellectual disability are more likely to suffer from cancer than those without intellectual disability (ID) using registry data from Sweden. The authors additionally perform analyses to assess whether the association is likely to relate to ID in a causal sense.

1) Due to the design (the oldest individuals in the analysis are 43 years old, and the vast majority of years-at-risk are in much younger individuals), this analysis focuses on cancers of young and early-middle age. This should be clearer in the manuscript. For example, the title could state: "Cancer risk in individuals with intellectual disability: analysis of registry data on young and early-middle age Swedish participants" (or similar).

2) Is the definition of syndromic ID standard? It seems somewhat narrow to my untrained eye.

3) While I understand that some of the motivation for the investigation comes from the desire to analyse rare cancers, some of these cancers are very rare. Aside from the wide confidence intervals in some analyses, this also leads to an upward bias in estimates that results from only including cancers with at least one case. If a cancer had no cases amongst individuals with ID, then this cancer would not be included in the analysis. The restriction to cancers with at least one case means that for very rare cancers, a positive hazard ratio (HR) is guaranteed. Additionally, as an individual can contribute to more than one cancer, it may be that several of the positive HR estimates are due to a single individual with multiple rare cancers - while the wide confidence intervals indicate imprecise estimates, they aren't able to inform the reader whether these are multiple independent wide estimates, or multiple dependent wide estimates taking positive values due to one or two multimorbid individuals.

For this reason, I would suggest either dropping the analyses of cancers with small numbers of cases in individuals with ID (eg only cancers with >5 cases), or greater aggregation of cancer outcomes, to include cancer types with no cases in individuals with ID - this is important to ensure that estimates are not biased by the exclusion of cancer types with zero cases. Greater aggregation of cancer outcomes would also ensure that individuals with multiple cancer types do not have an undue influence on analyses.

4) Page 7: "To adjust for potential confounding or effect modification..." - does covariate adjustment account for effect modification?

5) The list of covariates on page 7 is somewhat confusing. My initial reading was that these listed covariates would be adjusted for in the analysis. But on closer reading, these variables are only adjusted for in supplementary analyses? And it's not clear from eTable 8 that there is any analysis performed that adjusted for all of these variables? - you stratify on some of them, but I think you only adjust for at most one of these at a time?

I understand that the main analyses should be careful not to over-adjust (as potentially some of these variables are mediators), so for me, this is a case of clarifying the text rather than changing the analysis.

6) Page 8: "Attained age was used as the underlying time scale" - what is attained age?

7) Page 9: "We examined the effect of intelligence quotient (IQ) score on cancer risk through deriving IQ score from the ICD diagnostic codes." - please be careful of causal language.

8) Tables 2-4 may be clearer as figures. It's difficult to take in and compare all the HRs - would be more impactful if the reader could scan a column of visual estimates rather than have to read all the numbers.

9) Would be good to include the headline results from the supplementary and sensitivity analyses in the manuscript (eg the HR for all-cancer from the sibling comparision, the HR for the association between IQ and cancer risk, etc).

10) Page 11: "By visual inspection (eFigure 2), cancer risk appears to increase with age for cancer in the esophagus, stomach, small intestine, pancreas and other or unspecific sites among individuals with ID, compared with the reference group" - it's not clear what comparison you are making here.

11) eTable 7: Two of the odds ratios are over 900 million - something not quite right here.

12) Discussion: Is reverse causation a possibility, particularly for CNS cancers? Could be that pre-clinical cancer increases risk of ID?

13) Abstract: The claim of "detailed adjustment for potential confounding" is somewhat misplaced (in the text, the claim is phrased as: "adjusted for a rich set of potential confounders"). First, it's not clear why the authors want to adjust for large numbers of confounders - if this analysis is simply performed to identify whether individuals with ID are at increased cancer risk, then adjustment for covariates is unnecessary. Secondly, the main model 1 and 2 do not adjust for large numbers of covariates (again, I don't think this is unreasonable). Thirdly, many key covariates (BMI, smoking status, alcohol status, etc) are not available in the dataset. And finally, it's not clear that all the covariates are confounders. So while I value the potential for addressing confounding in the within-family analyses (and would suggest that eTable 6 may be worth including as a main table/figure), I wouldn't say that comprehensive covariate adjustment is a feature of this work (not should it be).

Overall, the researchers provide convincing evidence that individuals with ID are at greater risk of cancer both when comparing to the general population, and when restricting to matched comparisons within family units. I am cautious about the validity of some of the detailed comparisons of cancer subtypes, and would encourage either focus on a smaller number of cancer subtypes, or aggregation of related subtypes. However, I leave the final decision here to subject-matter experts.

*** Reviewer #4: 

This is a well-structured and well-written article with well-defined title, clear aim. The important finding of this study is the 1.6-fold increase of cancer rate among patient with ID. Regarding methodology, the process of subject selection was clear, variables were defined and measured appropriately, and study methods seemed to be valid and reliable. But I like to request for further checking of methodology by more experienced one. This study also highlighted some possible mechanisms behind this scene, which made it praiseworthy and helpful for further research. From my side, there is no correction needed. 

*** Reviewer #5: 

This manuscript by Liu et al is one of the largest longitudinal studies that has been carried out to estimate the risk of cancer among individuals with intellectual disability. The study shows that individuals with ID have an increased risk of cancer and suggest extended surveillance/early intervention in cancer. 

The manuscript is well written, addresses the strengths and weakness of the study and compares with previous studies. I recommend this manuscript for publication to advance the understanding in this field. Following are some of the points that I would like to get some clarifications on:

1. In one of the previous studies by Sullivan et al. 2004 (PMID: 15801486), males with ID were observed to have a significantly increased risk of leukemia, brain and stomach cancers and a reduced risk of prostate cancer, while leukemia, corpus uteri and colorectal cancers were significantly higher in females. Did the authors in the current manuscript investigate it? Authors mention on Page 11 that 'The increased risk of any cancer did not vary by ID severity or between men and women….". Authors should include in the discussion why they do not see the increased risk of cancer among men and women in comparison to Sullivan et al. 

2. eTable5 shows characteristics of individuals with ID by severity of ID. The percentage of different categories of ID were shown for males. Is there a reason why this percentage were not shown for females? Authors should show the percentage of females with breakdown into different ID groups as they have done for males in this Table to make it complete. Looks like out of 27,956 individuals with ID, about 16,221 (58%) were males and about 11,735 (~42%) of them were females.

3. Authors also mention that compared with the reference group, individuals with ID were in general more likely to be male. They should mention in the manuscript the possible reason behind it. We know there is a higher prevalence of ID in males than in females.

***

[LINK]

---

## [Decision Letter · Decision Letter 2]

30 Sep 2021

Dear Dr. Liu,

Thank you very much for re-submitting your manuscript "Cancer risk in individuals with intellectual disability: A population-based cohort study" (PMEDICINE-D-21-01891R2) for consideration at PLOS Medicine.

I have discussed the paper with our academic editor and it was also seen again by four reviewers. I am pleased to tell you that, provided the remaining editorial and production issues are fully dealt with, we expect to be able to accept the paper for publication in the journal.

[LINK]

Please let me know if you have any questions, and we look forward to receiving the revised manuscript.   

Sincerely,

Richard Turner, PhD

rturner@plos.org

Requests from Editors:

We suggest adding "in Sweden", or similar, to the title. 

At line 36 (abstract), prior to the presentation of the quantitative findings, we suggest adding a sentence or two to provide some additional detail: including the higher rate of ID in males, the length of follow-up, and the numbers of cancers in both groups.

At line 38, please make that "the esophagus ...".

At line 46, please remove "Despite the large sample size and high-quality data ..."; we suggest adapting the text immediately after this point to: "The main study limitations were the limited statistical power for the analysis of specific cancer types, and the potential for underestimation of the studied associations (e.g., due to potential under-detection or delayed diagnosis of cancer among individuals with ID).".

At line 51, please adapt the text to the style "In this study, we found that ...".

Please add bullets to the individual points in the Author summary. 

Please also expand the Author summary so that each subsection has 2-3 points. For example, in the final subsection an additional point could cite a need for further research on incidence of some specific types of cancer.

Please remove the word "prospective" at line 282 and any other instances (we view this as a retrospective analysis). 

Please use the general style "age 43 years" throughout. 

Noting reference 7, some punctuation should be removed from the author list (to leave "Linn JG, ..."); in reference 45, authors' initials should be used. 

We suggest moving S1 Figure to the main part of the paper. 

Please relabel the STROBE attachment "S3_STROBE_Checklist" and refer to it by this label in the Methods section. 

Comments from Reviewers:

*** Reviewer #1: 

This article is definitely among the largest of its kind in terms of sample size. The current form of the article seems streamlined and easier to read. The amendments made in this article are commendable. I would recommend the article to be published in its current form.

The conception of idea for the research as well as its novel finding will definitely be of important concern in the future. It might have a public health impact in the sense that recommendations about screening for malignancies for cases with intellectual disabilities could be based on these findings.

I would like to congratulate the whole team for their efforts.

Thank you.

*** Reviewer #2: 

Again, I believe this research extends existing knowledge well and is impactful due to its large sample and thorough methodology. The authors address all my points appropriately and I believe this article is now clear and strong. 

I will allow comments on changes to the statistics to be made by reviewer 3, but I have a few other minor points outlined below. After these points have been addressed, I am satisfied and happy for this work to be published in PLOS medicine. 

1) P.8 Line 154-6, I feel this sentence could be a little clearer: "An individual with more than one cancer type (one individual with ID and 422 individuals without ID) was counted as a cancer case in the analysis of several cancer types"

a. It is unclear if this means they were put into a completely separate analysis. A little clarity would help. 

2) Great to see that the authors were responsive to my comments on sex and gender. One last addition to this point, on P.8 Line 158, perhaps add "(male/female)" after the variable "sex" so that the reader knows that intersex was not included in this variable. 

3) P.10 Line 198, there seems to be a spelling error: "within-sibship"

4) P13 Line 276, possibly another spelling error: "statistically significantly" should be "statistically significant"

5) P14 Line 294, the adjustment for potential confounders should be considered in relation to comments from reviewer 3, however, maybe at this point you could explain why finding that confounders did not impact results is important? 

6) P17 Line 348, you abbreviate ALL but not AML, is there a reason for this?

7) Double check gramma in the discussion, I think a few changes have been made that the authors should give another once over to improve readability. 

*** Reviewer #3: 

Thanks to the authors for their comprehensive revision of this work. It is reassuring to know that the bias arising from considering cancers with at least 1 case in individuals with ID is not considerable. It is also reassuring to know that the results for rare cancers are not driven by a single individual (or a small number of individuals) with ID who are massively multimorbid.

I have a few comments:

Paragraph on Supplementary and sensitivity analyses - if there are key summary measures or estimates, I'd consider providing these in the manuscript.

Figure 1: For me, this is hard to read, and it's a shame to lose the uncertainty intervals for the estimates. I would suggest a single vertical axis, and paired estimates plotted with the uncertainty interval horizontally - one colour for syndromic, another colour for idiopathic. Similar to a forest plot, but with equal sized points and no pooled estimate. I think a series of paired points and horizontal lines (with estimates along the right hand side?) would provide a more compelling presentation. From a quick search, this is close to what I had in mind: https://plos.figshare.com/articles/figure/_Forest_Plot_of_Unadjusted_and_Adjusted_Hazard_Ratios_for_Any_and_Violent_Convictions_/1482255/1 - although with pairs of points, not triples, and using colour. (Just to clarify, this is a suggestion - the authors should decide the optimal way to present their data in consultation with the editors.)

If the authors do make this change, I'd consider similar changes for Supp Figures 2 and 3.

S2 Table - what are the %s? Are they % of the overall population with this cancer? Or % of all cancers?

S6 Table - the inexact phrasing "effect of IQ score" is still in the heading.

---

Stephen Burgess

*** Reviewer #5: 

The manuscript is considerably improved from the first submission. I accept it in its current form.

***

[LINK]

---

## [Editor Report · Decision Letter 3]

8 Oct 2021

Dear Dr Liu, 

On behalf of my colleagues and the Academic Editor, Dr Zheng, I am pleased to inform you that we have agreed to publish your manuscript "Cancer risk in individuals with intellectual disability: A population-based cohort study in Sweden" (PMEDICINE-D-21-01891R3) in PLOS Medicine.

Prior to final acceptance, we suggest moving "in Sweden" before the colon (title). 

PRESS

Sincerely, 

Richard Turner, PhD 

rturner@plos.org